# Executive Functioning in Adults with Down Syndrome: Machine-Learning-Based Prediction of Inhibitory Capacity

**DOI:** 10.3390/ijerph182010785

**Published:** 2021-10-14

**Authors:** Mario Fernando Jojoa-Acosta, Sara Signo-Miguel, Maria Begoña Garcia-Zapirain, Mercè Gimeno-Santos, Amaia Méndez-Zorrilla, Chandan J. Vaidya, Marta Molins-Sauri, Myriam Guerra-Balic, Olga Bruna-Rabassa

**Affiliations:** 1eVIDA—Lab, Faculty of Engineering, Deusto University, 48007 Bilbao, Spain; mariojojoa@deusto.es (M.F.J.-A.); mbgarciazapi@deusto.es (M.B.G.-Z.); amaia.mendez@deusto.es (A.M.-Z.); 2Faculty of Psychology, Education and Sports Sciences Blanquerna, Ramon Llull University, 08022 Barcelona, Spain; sarasm0@blanquerna.url.edu (S.S.-M.); mercegimenos@blanquerna.url.edu (M.G.-S.); MartaMS6@blanquerna.url.edu (M.M.-S.); myriamgb@blanquerna.url.edu (M.G.-B.); 3Department of Psychology, Georgetown University, Washington, DC 20057, USA; cjv2@georgetown.edu

**Keywords:** aging, artificial intelligence, cognition, Down syndrome, executive functions, feature selection, inhibition, machine learning, neuropsychology

## Abstract

The study of executive function decline in adults with Down syndrome (DS) is important, because it supports independent functioning in real-world settings. Inhibitory control is posited to be essential for self-regulation and adaptation to daily life activities. However, cognitive domains that most predict the capacity for inhibition in adults with DS have not been identified. The aim of this study was to identify cognitive domains that predict the capacity for inhibition, using novel data-driven techniques in a sample of adults with DS (*n* = 188; 49.47% men; 33.6 ± 8.8 years old), with low and moderate levels of intellectual disability. Neuropsychological tests, including assessment of memory, attention, language, executive functions, and praxis, were submitted to Random Forest, support vector machine, and logistic regression algorithms for the purpose of predicting inhibition capacity, assessed with the Cats-and-Dogs test. Convergent results from the three algorithms show that the best predictors for inhibition capacity were constructive praxis, verbal memory, immediate memory, planning, and written verbal comprehension. These results suggest the minimum set of neuropsychological assessments and potential intervention targets for individuals with DS and ID, which may optimize potential for independent living.

## 1. Introduction

Longer life expectancy in people with Down syndrome (DS), often into the fifties, and high rates of cognitive decline have led to a growing interest in the study of their aging process [1,2,3,4,5,6,7]. and in ways to encourage healthy aging [8,9,10,11]. Patterns of cognitive changes during aging in adults with DS are diverse [12,13] but most severely affect memory, language, visuoconstructional skills, executive functions, and motor praxis skills [14,15,16,17,18,19,20]. Furthermore, a greater predisposition to develop Alzheimer’s disease (AD) [21,22,23,24] has been noted, with onset of dementia marked by declines in episodic memory, visuospatial organization, visuospatial memory, and executive functions [25]. Importantly, alteration in executive functions has been recognized as one of the first symptoms of AD in persons with DS. Therefore, assessment and understanding of these functions in adults with DS is critical for early diagnosis of AD and a better quality of life [26]. Executive function (EF) is a multidimensional construct that includes self-regulatory processes of response inhibition, working memory, and cognitive flexibility [27]. Among the general population, good executive function performance in the early stages of life predicts positive outcomes in adulthood, including better social relationships, higher levels of employment, and fewer risky behaviors [28,29]. Together with other cognitive functions, response inhibition in particular has been singled out as a process that promotes cognitive performance that is adaptive to one’s environment [30]. In the adult population with DS, changes in executive functions (EF) are characterized by difficulties with working memory, sustained attention, planning, and inhibition [31]. An age-related decline in executive functions has also been observed in connection with verbal working memory and visuospatial planning such that a decline in inhibition capacity and cognitive flexibility precedes that in verbal/visuo-spatial memory [32,33,34,35,36]. Improvement has been noted in the affected executive functions such as working memory, inhibition, and planning after using neuropsychological intervention programs [26]. Longitudinal studies in adolescents with DS and intellectual disability (ID) indicate significant changes in planning tasks, verbal fluency, and short-term memory in 3-to-5-year longitudinal monitoring. However, persons with ID are distinctive for inhibition control deficits [32,33,37]. Difficulties are evident in tasks that require simultaneous attention, short-term verbal memory, and working memory, leading to poorer performance in adults over 50 [37]. Other studies focusing on persons with DS reveal an age-related decline in verbal working and visuospatial memory in addition to mental flexibility [34,35,38]. They also suggest a greater decline in comparison to persons with ID due to other etiologies in planning, attention, and verbal fluency [39,40]. Verbal intrusions associated with executive functions have been observed in middle-aged adults with DS [31,32,33]. A positive correlation between inhibition capacity (measured by the Cats-and-Dogs test) and visual memory (measured by object memory) has been observed. Other studies have shown a significant correlation between age and planning capacity, long-term selective memory and receptive language [41]. Thus, executive function impairment co-occurs with a decline in memory skills in persons with DS. Autonomy and quality of life factors have increased for persons with DS in recent years as a result of the support they have received in adapting to the working world. However, fewer persons with DS are hired due to their EF deficits [42]. Understanding and training EF performance could increase the quality of life for people with DS and also improve patterns related to inhibition control and self-regulation [26,43]. Different studies suggest that practical skills, referred to as daily tasks, are a strength for people with DS since they often remain stable as they age [43]. However, their EF deficits cause them to experience difficulty in successfully managing certain daily situations [43]. Furthermore, when persons with DS have dementia caused by AD, they undergo a marked decline in their overall functioning because it is a progressive condition and therefore seriously impacts their quality of life. For this reason, it is important to find out more about EF in persons with DS in order to improve their quality of life and autonomy [26]. In summary, further study is needed on EF decline in aging adults with DS [44], considering that the role of these higher cognitive functions is to control and regulate cognitive skills. It has also been suggested that early clinical detection of AD in persons with DS coincides with executive deficits. The results indicate that the general population manifests short-term memory decline as the most common indicator associated with AD onset, whereas persons with DS normally show executive dysfunction and behavioral and psychological symptoms in the preclinical stages, which may precede loss of memory [45]. Furthermore, it has been suggested that one indicator of AD in persons with DS is loss of daily living skills [46] and that they are more affected as they grow older [47]. On the one hand, boosting their cognitive reserve is vital. This may be achieved through interventions in their school and work environments and could help to reduce the cognitive decline associated with dementia [48,49]. In light of the importance of executive function to adaptive outcomes and their co-occurrence with memory and other cognitive deficits, we aimed to identify cognitive factors that predict inhibition capacity in adults with DS. The application of machine learning algorithms was chosen for this study, as they are widely used to identify the role of features and/or variables when searching for specific behavior of a certain phenomenon. This is generally achieved through training and testing supervised predictive models that recognize relational patterns between a response variable and some input variables. As a result, the necessary procedure focuses on training three predictors, and based on their implicit metrics, determining which of all the input variables involved have the greatest impact on the behavior of the dependent variable (inhibition capacity). The algorithms used are Random Forest, logistic regression [50], and support vector machines, and they were selected for their widespread scientific use in this type of application. Furthermore, the intersection of sets is proposed as the ensemble method for the results obtained by each algorithm, as well as weight-based organization in order to normalize the results between the three models used.

The research objective is to apply artificial intelligence techniques to analyze the cognitive performance results collected in previous studies on the adult population with DS in order to identify the variables with the greatest impact on improving cognitive performance in everyday activities in the aging process.

## 2. Materials and Methods

### 2.1. Materials: Data Description

The data analyzed were obtained from a cross-sectional study [51]. The participants were recruited through intentional non-probabilistic sampling. The sample consisted of 188 adults with DS who evidenced mild or moderate ID. The participants were recruited from 26 Spanish and South American institutions and foundations for the care of persons with ID. They agreed to participate in this study on a voluntary basis. The confidentiality of the data for the present study has been preserved at all times. Family members and the participants themselves have signed informed consent forms, in compliance with Organic Law 15/1999 on Personal Data Protection and the application of Law 41/2002 on Basic Regulation of Patient Autonomy and Rights and Obligations in terms of information and clinical documentation. Furthermore, article 27 of the Helsinki Declaration [52] has been observed. It regulates the criteria for the publication of research results.

Inclusion criteria for the sample are being an adult (over 18 years of age) with DS, of either sex, and that the participants and their legal representatives have signed the informed consent form. Persons with a diagnosis of a neurological, neurodegenerative, or severe mental illness that could significantly interfere with the test results were not eligible to participate. Persons who had been diagnosed with non-disability-related physical and/or cognitive changes and had severe sensory impairment were also excluded.

The ages of the study participants range from 19 to 62 years, with an average age of 33.63 (SD = 8.81). The sample includes 93 (49.47%) men and 95 (50.53%) women. 

### 2.2. Research Instruments

Overall cognitive performance assessment of the participants is particularly important due to the wide diversity of cognitive skills in the population with DS. Raven’s Colored Progressive Matrices test was used for the assessment of intellectual function [53]. 

Cognitive assessment included 21 tests selected from different neuropsychological batteries that have been validated for the Spanish population, in addition to ad hoc measures created for assessing orientation to time, place, or person. As the population under study was formed by persons with DS, experts were consulted to obtain evidence of the validity, concordance, and relevance of the cognitive assessment tests based on their content. The set of 21 tests was satisfactorily validated for content by the experts [51]. Table 1 shows the selected tests for assessment of the following cognitive domains: memory, attention, language and communication, executive functions, and praxis. 

The Cats-and-Dogs test, from the Cambridge Executive Functioning Assessment battery (CEFA) [31], was administered to assess response inhibition capacity. In the test, participants were asked to indicate images of a “cat” and a “dog”. They were then asked to say “cat” when they were shown an image of a dog and vice versa. They received points for good performance in the second stage of the test. The values that participants could obtain ranged between 0 and 5, according to the response execution level. 

The database consists of a table with 39 variables and 188 registers. Some fields contain missing data, labelled 999 or null. We show the number of values to recover in Table 2.

In the same way, the procedure to convert the variable Cats-and-Dogs from an initial polychotomous domain to a dichotomous domain was to assign zero (0) to the values 0, 1, and 2 and one (1) to the values 3, 4, and 5, since it is considered that between 0–2 the performance is worse than from a value of 3. This gives a binary response variable, whose behavior is determined by the binomial distribution. 

### 2.3. Methods

This work proposes a machine-learning-based system for the automatic selection of the main attributes that best explain the Cats-and-Dogs response variable (EF). This variable plays a key role in the psychological scope for the study of cognitive disability in humans. Hence, a system composed mainly of three stages is proposed. Initially, it allows the recovery of the missing data in the database, removal of correlated variables and automatic selection of the most important variables. The following block diagram shows the proposed stages (Figure 1).

Machine learning is not the only approach that may be applied to these data in the study. We selected the approach that we thought was best suited to our multivariate dataset, because it is data driven and requires few decisions for selecting potentially predictive variables.

#### 2.3.1. Support Vector Machine as the Recovery Algorithm to Retrieve the Missing Data

Data recovery uses related and independent data to recover the missing values with a machine learning algorithm [62]. For this task, a support vector machine was trained to carry out a regression [63], selecting the response variable as the one whose data were to be recovered, and the rest of the variables (with no missing data) as input variables. To measure the performance of the model for the data recovery task, we used the mean absolute percentage error [64] (MAPE), the mathematical expression of which is shown in Equation (1):(1)$$\mathrm{MAPE}={\frac{\Sigma \;\frac{\left(real_{i\;}-Pred_{i}\right)}{real_{i}}}{n}}_{}\;*100\%$$

#### 2.3.2. Dimensionality Reduction Based on the Pearson Correlation Coefficient

Based on [65], it is required that the input variables do not show correlation in a machine learning analysis in order to suppress disturbance and interference between them. Hence, we have obtained a Pearson correlation matrix per each field in order to select the variables that exceed one umbral. The following block diagram contains a description of each of the stages of the procedure (Figure 2).

##### Correlation Matrix Generation with All Variables

We built a Pearson correlation with all variables using the data described in Section 2.1. The purpose was to carry out a quantitative observation of the correlation between pairs of variables. This information was used to remove the redundant information in the training set of the machine learning models, in order to improve their performance. 

##### Fixing the Pearson Correlation Coefficient Threshold 

After discussion with experts, we proposed three threshold values, 0.7, 0.8, and 0.9, respectively, to obtain subsets of variables to be analyzed independently.

##### Highlighting the Correlated Variables

The correlated variables could be removed once their absolute Pearson correlation exceeded the threshold set by the researchers. Therefore, highlighting them facilitates the task of the selection and removal of variables, based on the researchers’ knowledge of psychology.

##### Variable Elimination by Experts

The variables marked in the previous step were carefully revised by the experts, mainly taking into account their psychological significance. Once they were studied, the ones that should be removed to reduce the size of the dataset in terms of number of variables were identified.

##### Generation of New Databases to Perform the Analysis

Finally, three datasets were obtained. These datasets were the ones used for the automatic selection of variables, applying the Random Forest, logistic regression, and support vector machine algorithms, as described in the following subsections. 

#### 2.3.3. Automatic Variable Selection Algorithms

The variable selection algorithms applied were based on three metrics which are widely used in the state of the art: entropy, logarithmic probability, and margin optimization coefficients. Each of them are described below:

##### Random Forest as the Variable Selection Algorithm (RF)

The Random Forest is an ensemble of decision trees [66] through the technique called bagging, in order to increase the generalization capacity and decrease the variance for the desired performance metrics, with the purpose to select, in an objective manner, the variables that impact the prediction of the output Cats-and-Dogs. This approach works by calculating the entropy [67] of the data for each tree and using them to determine the variables that provide the most classification information.
(2)$$Entropy=-p_{i}*log\left(p_{i}\right)$$

The bagging [67] classifier proposes a meta-algorithm to combine machine learning algorithms in order to improve the metrics of the overall performance of the system used. The following block diagram shows the model used (Figure 3).

Each DTnblock corresponds to a decision tree trained with an independent part of the data, and their ensemble is performed in the last block, bagging in inference time to give a consensuated output. 

##### Logistic Regression as the Variable Selector Algorithm (LR)

Logistic regression was used for this study, as the intention was to predict the Cats-and-Dogs variable, and determine which variables interact most for its prediction. 

It works by analyzing the output under a binomial distribution, as shown hereafter [68]
(3)$$Y\;∼B\left(n_{i.\;}p_{i}\right)\;$$
where *n_i_* corresponds to the number of Bernoulli trials and *p_i_* to the known probabilities. This enables us to obtain a list of logarithmic probabilities or Logits [68]
(4)$$\frac{p_{i}}{1-p_{i}}=e^{\left(b_{0}+b_{1}x_{1}+....+b_{j}x_{j}\right);\;Logit_{\left(Y_{i}\right)}=log\;\left(\frac{p_{i}}{1-p_{i}}\right)}$$

As can be seen in Equation (4), the incidence of *X_i_* array, of size *j* variables, is represented as the logarithmic probabilities related to the occurrence of the output variable *Y_i_*, which is dichotomous for this case.

##### Support Vector Machines as the Variable Selection Algorithm (SVM)

The operation of the support vector machine is based on the margin maximization (distance between the support vectors and the data used) in order to draw a hyperplane that represents the training phase of the algorithm. In inference, the relative position of the individual vectors is compared to the hyperplane, and it is used to define the degree of membership to the set classes. Once the algorithm has been trained, it is possible to access the model weights [68]
(5)$$H=W^{T}*X+B$$

*W^T^* is an array of vectors whose direction focuses towards the desired solution.

The importance of the characteristic can therefore be determined by comparing the size of these coefficients with each other. Hence, it is possible to identify the main characteristics used in the classification by observing the SVM coefficients and removing the ones that are not important (that have less variance).

Finally, the reduction in the number of variables in machine learning plays a key role, particularly when working with large datasets. In fact, it can accelerate training, avoid overfitting, and ultimately lead to better classification results thanks to noise suppression in the data. 

##### Feature Selection Committee of All the Algorithms

Each of the described algorithms will return in its output the variables considered as the most important in relation to its operation metrics for the prediction of the Cat-and-Dog variable (EF). For this reason, the decision was made by selecting the common variables in the output of the three machine learning algorithms applied, in order to detect and mitigate the implicit algorithmic bias. 

##### Feature Selection by Importance Index

Similar to the previous stage, the results of the three algorithms were taken into consideration to decide which variables were most important in predicting the Cat-and-Dog response variable. However, at this stage, the selection principle was different. Each algorithm gave us a score depending on the order of the output of the variables. This was performed to normalize the results in regard to the internal execution metrics of the machine learning algorithms applied. It is important to mention that the first 20 outputs of each algorithm were chosen strictly by order. Therefore, the score assigned ranged from 1 to 20, with 1 point for the last variable and 20 points for the first output variable. Finally, the ones with the highest number of points were selected.

## 3. Results

### 3.1. Data Recovery

Table 3 shows the recovered data available for use when applying machine learning techniques.

Table 4 shows the errors obtained for the data recovery problem, for each of the variables to be recovered. The available data were split into 80% for training and 20% for testing to elaborate the following table.

Once the data recovery algorithms had been applied, the correlated variables were removed to avoid redundant information and thus improve the performance of the proposed machine learning models. 

### 3.2. Removal of Correlated Variables

The removal of variables for thresholds 0.9, 0.8, and 0.7 are shown below (Table 5).

It is important to highlight that the number of removed variables is cumulative. In other words, the variables that correlated with lower Pearson threshold correlation coefficients are also removed from higher thresholds. 

### 3.3. Committee- or Intersection-Based Variable Selection

The RF, LR, and SVM machine learning algorithms described above for the automatic selection of variables were run for each of the subsets obtained, with the variable removal criteria described in Section 3.2. Furthermore, the intersection was taken as the final result to reach consensus on the total results of all the algorithms. This result refers to the output variables common to the three machine learning techniques. Table 6 shows the results obtained.

Each column of Table 6 shows the most important variables for each subset obtained after variable removal. These are the variables that most influence prediction of the EF variable (Cats-and-Dogs). In other words, they are the most important in explaining the value obtained in the response variable. We have selected the column corresponding to variables with <0.8 Pearson correlation, based on our experience and knowledge about the focus study area, taking into account that more cognitive functions are represented in this column. Therefore, the best variable predictors of inhibition capacity were functions related to praxis, memory, attention, executive functions, and language: oral verbal comprehension, verbal memory, overall mental control, direct digits, constructive praxis, errors of memory of images, and written verbal comprehension.

### 3.4. Variable Selection Based on the Importance Index

As in the above table, the decision was made to use the results of the three machine learning algorithms together; however, with the difference of assigning each feature a weight on a scale of 1 to 20 according to the importance it is given in each algorithm. In this way, the result of the importance given by each algorithm is consistent, as they use different metrics. For instance, we can observe two with a significant weight for individual algorithms and not just the intersection of the overall results. Table 7 shows the results obtained.

Regarding the weight of each attribute, the cognitive variables that best predict the inhibition capacity are first the constructive praxis, followed by verbal memory, verbal written comprehension, general cognitive performance, direct digits, response speed to inhibition capacity, orientation in person, memory of visual memories, age, vocabulary, denomination of images, orientation in place, clock test copy, and verbal fluency. Thus, these cognitive variables which best predict the inhibition capacity are related to praxis, memory, attention, executive functions, and language.

### 3.5. Performance of the Algorithms When Classifying the EF Variable

In order to observe and compare the performance of the machine learning models used in the classification of the binary variable EF (Cats-and-Dogs), the decision was made to train and test each of them. The data were split into 80% and 20%, respectively, for the subsets obtained after the removal of variables (Table 8).

As shown in Table 8, the model with the best performance was Random Forest, with a balanced accuracy of 86% for the dataset formed by all the variables, except those that show a Pearson correlation above the 0.8 threshold (metrics highlighted in bold font). It is also highlighted that the balanced accuracy metric was used for this application, so the dataset shows a considerable imbalance of 86%, as regards the amount of data per class of the Cats-and-Dogs response variable.

## 4. Discussion

The results of the present study show that three machine learning algorithms convergently indicated that the best predictors of inhibition capacity, an executive function, in our population with DS spanning young adulthood and aging, were the cognitive domains of praxis, memory, attention, planning, and language.

Past work has established that response inhibition is a key element of executive function that helps people with DS adjust to the demands of their environment. Cognitive changes during aging in persons with DS are characterized by difficulties in working memory, sustained attention, planning, and inhibition capacity [69], and it is particularly important to conduct further study on their relation to other cognitive domains [44]. A deeper understanding of executive function performance in adults with DS, and of the factors that predict it, is necessary to develop training regimens that could improve inhibition capacity and self regulation, which is necessary to increase their quality of life [26,30,43].

This study focused on determining how the neuropsychological assessment of cognitive performance during aging in persons with DS can help to predict inhibitory control capacity. The results demonstrate that, among the cognitive functions assessed in this population, the best predictors of inhibition performance, measured through the Cats-and-Dogs test, are constructive praxis, verbal memory, immediate memory, planning, and written verbal comprehension. This set of cognitive functions is also known to be most vulnerable to aging in persons with DS [15,16,17,18,19,20,70].

Our results reveal some important insights that are useful for informing which functions should be the focus of the early detection of age-related impairment and intervention strategies for people with DS. First, constructive praxis was the cognitive function that carried the most weight in the prediction of inhibitory control capacity. Along with perceptual–motor functioning, performance in constructional praxis tests requires the ability to plan and organize actions, which draws upon processes of executive function. Indeed, recent studies showed that both praxis and inhibitory capacity decline in aging [51,71]. Second, verbal short-term memory predicted inhibitory function. In previous studies, short-term memory decline was found to be more frequent in adults over 35, showing an earlier onset and faster progression [72,73,74]. Further, immediate memory, assessed with the Digit Span test, also predicted inhibition capacity. Short-term and immediate memory are good indices of attention, which is sensitive to age-related changes in adults with DS. It is important to note that sleep can affect attentional performance, and that such deficits could be related to sleep apnea, which is frequent in DS [26,75]. Third, planning, as measured by a clock drawing test, and written verbal comprehension were predictors, although with lower weights than praxis or memory. Both tests require some motor planning similar to constructional praxis, which was highly predictive of inhibition capacity. Verbal ability per se appears to be relatively stable in adults with DS until 40, 50, or even 60 years of age. The decline in verbal skills seems similar to that of the elderly of the general population [76], although it appears to occur earlier in persons with DS [40,77,78]. Together, our results suggest that these cognitive functions, which reduced executive performance, could be the minimum set of neuropsychological assessment factors needed for the early detection of AD in adults with DS, and could be used as intervention targets for improving executive function performance [26]. 

Assessment is therefore vital to determining the progression of age-related cognitive deterioration. 

The studies reviewed from the literature indicate that there is still relatively little research with conscientious follow-up on the cognitive changes in adults with DS and, more concretely, of their executive functions, as this assessment is a complex process. Nevertheless, we know that persons with DS show inhibitory control deficits, which makes it vital to gain a better knowledge of how to assess these types of functions in order to improve their quality of life [32,33,37]. In spite of having made progress in recent years, finding appropriate instruments adapted to their cognitive profile is a challenge and a barrier to the early detection of the cognitive changes associated with the onset of AD. The assessment of poorer performance in executive functions is key to achieving early diagnosis, since recent studies show that early clinical onset of AD coincides with frontal symptoms and may precede memory loss in persons with DS [45]. Furthermore, inhibition capacity is vital to boosting cognitive performance adapted to one’s environment and is a key element in self-regulation, together with other cognitive functions such as attention, memory, or communication capacity. These mechanisms are vital to functioning successfully in one’s environment, as they enable the creation of strategies and resources for self-management [30].

Some limitations of this study must be noted. First, the lack of adequate information on each participant’s degree of disability prevented accurate determination of the relationship between cognitive changes and autonomy in daily life, taking into account their level of dependency. Second, it is worth emphasizing that administering neuropsychological tests adapted to adults with DS for precise assessment of all their cognitive functions is difficult. Although considerable progress has been made in recent years in designing standardized neuropsychological assessment adapted to the adult population with DS, more progress needs to be made in the accurate evaluation of the changes in different cognitive domains during the aging process. Third, it would have been useful to include more measures of executive functions, apart from inhibition capacity. We did not include some measures (e.g., reverse digits) because of a floor effect. Fourth, the machine learning algorithms used to identify key factors assume a correlation between the response variable and the training variables. This relationship may be linear or nonlinear. Nonlinear algorithms such as Random Forest and logistic regression were used to ensure better results. However, for the support vector machine algorithm, a linear relationship was assumed, which may have biased the results. Lastly, the amount of available data for this study only allows analysis with traditional machine learning algorithms. This limitation thus raises a future line of research that requires gathering more data. In this manner, with a sufficiently large amount of data, deep-learning-based classification algorithms could be used, which would make the results more accurate.

## 5. Conclusions

Based on the results of this study, we can conclude that the application of artificial intelligence techniques to cognitive assessment data identified that the best predictors of inhibition capacity during aging in adults with DS were praxis, memory, attention, motor planning, and written language. In light of the importance of inhibition capacity for adapting to one’s environment and optimizing performance in daily life activities, we suggest that these functions should be included in neuropsychological assessment protocols for adults with DS. Additionally, taking into account the importance of executive functioning in the early diagnosis of cognitive decline associated with AD onset, knowledge of inhibition capacity performance will aid in such early detection. These results also suggest targets for the application of cognitive stimulation strategies to boost cognitive reserve and prevent age-related cognitive deterioration in persons with DS. Our results also demonstate the utility of classical artifical intelligence methods for exploring small datasets for the identification of important variables or features in comparison to classical statistical methods. 

## Figures and Tables

**Figure 1 ijerph-18-10785-f001:**

Block diagram of the proposed automatic variable selection model.

**Figure 2 ijerph-18-10785-f002:**
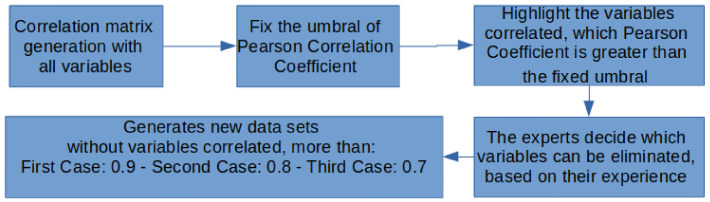
Block diagram of the steps to select the correlated variables.

**Figure 3 ijerph-18-10785-f003:**
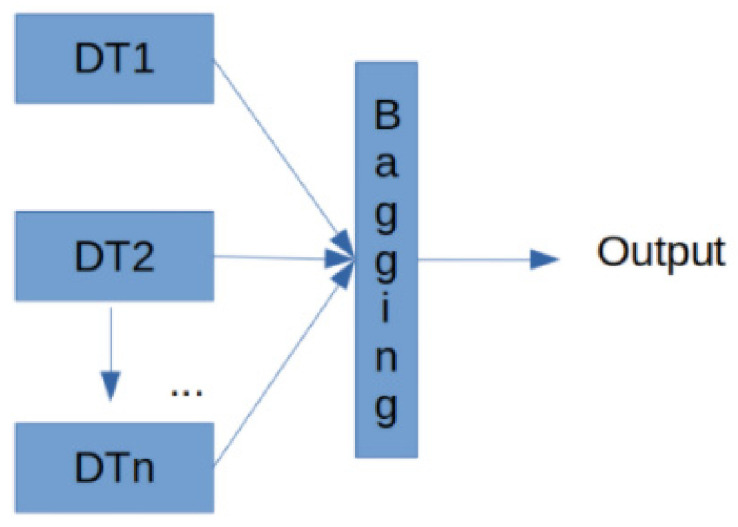
Model to detect the importance of the variables based on the decision trees used.

**Table 1 ijerph-18-10785-t001:** Neuropsychological tests to perform the cognitive assessment of adults with DS [51].

Cognitive Domains	Instruments	Acronyms of Variables Names
General cognitive performance	Scale color progressive matrices of RAVEN (RCPM) [53]	Raven
Memory (immediate, verbal memory, visual memory and visual recognition memory)	Memory of images (ad hoc)	Mem_ima
	Image recognition (ad hoc)	Mem_recog
	Verbal Memory 1a and 1b [54]	Mem_verbal
Attention (attention and verbal short-term memory)	Direct digits (K-ABC) [55]	Direct_D
Language and communication (receptive vocabulary, denomination, spontaneous language and verbal fluency)	Peabody Picture Vocabulary Test (PPVT) [56]	PPVT
	Visio-verbal denomination (ad hoc)	Total_denomin
	Spontaneous language: description of a sheet [57]	Spont_lang
	Verbal fluency: categorical evocation [58]	Verbal_flu
	Oral verbal comprehension (ad hoc)	OV_compr
	Written verbal comprehension [57]	WV_compr
Executive functions (executive function, processing speed, planning and motor execution)	Cats-and-Dogs test [31,59]	EF
	Clock test [60]	Clock_order // Clock_copy
	Motor execution 1	Mot_ex1
	Motor execution 2	Mot_ex2
	Overall motor execution [57]	Overall_ME
	Mental control—numbers	Mental_contr_num
	Mental control—days	Mental_contr_days
	Overall mental control [57]	Overall_mental_contr
Praxis (visio-constructive ability, imitation of postures, ability to imitate)	Constructive praxis [57] Imitation of bilateral postures [61] Ideational praxis [57] The last item has been replaced by another, more familiar and recognizable for the DS population.	Constr_praxis Imi_post Ide_praxis
Orientation (time, place, person)	Orientation in person (ad hoc)	OP
	Orientation in space (ad hoc)	OS
	Orientation in time (ad hoc)	OT
Writing	Graphics	Graphics

**Table 2 ijerph-18-10785-t002:** Description of the fields with missing values.

Variable Name	Amount of Missing Information
Spont_lang	1
Direct_D	2
Span	6
Deno_obj_body	1
Graphics	6
Mem_ima	1
Mem_recog	1
Errors	2
OV_comp	1
WV_comp	26
Mental_contr_num	1
Mental_contr_days	1
Overall_mental_contr	1
Mem_verbal	1
Imi_post	8
EF	4
Secs	5
Raven	3
Verbal_flu	3
PPVT	1
Clock_copy	3
ZEF	4
PE_EF	4
Category_EF	4
Category_EF_2	4
Z_secs_EF	5
PE_secs_EF_2	5
Categories_secs_EF	5
Categories_secs_EF_2	5

**Table 3 ijerph-18-10785-t003:** Data recovered from the database for machine learning analysis.

Description	Amount
Total complete data	7218
Total data recovered	114
Total data available for processing	7332
Total variables	39
Total registers	188

**Table 4 ijerph-18-10785-t004:** Mean absolute percentage error, MAPE, obtained for the recovery of missing data with the test dataset.

Variable Name	Amount of Recovered Data	Mean Absolute Percentage Error in Testing
Spont_lang	1	95.77%
Direct_D	2	98.61%
Span	6	99.06%
Deno_obj_body	1	95.27%
Graphics	6	99.68%
Mem_ima	1	96.3%
Mem_recog	1	99.9%
Errors	2	97.28%
OV_comp	1	97.62%
WV_comp	26	96.92%
Mental_contr_num	1	95.38%
Mental_contr_days	1	95.07%
Overall_mental_contr	1	96.67%
Mem_verbal	1	99.19%
Imi_post	8	98.56%
EF	4	95.61%
Secs	5	95.28%
Raven	3	97.44%
Verb_flu	3	96.49%
PPVT	1	95.13%
Clock_copy	3	96.84%
ZEF	4	98.69%
PE_EF	4	96.91%
Category_EF	4	96.38%
Category_EF_2	4	99.98%
Z_secs_EF	5	95.56%
PE_secs_EF_2	5	96.38%
Categorias_secs_EF	5	98.86%
Categories_secs_EF_2	5	95.2%

**Table 5 ijerph-18-10785-t005:** Removal of variables based on the Pearson correlation coefficient.

Threshold	Variable 1	Variable 2	Removed Variable
0.9	OT	OS	OT
	Mental_contr_num	Overall_mental_contr	Mental_contr_num
	Mental_contr_days	Overall_mental_contr	Menta_contr_daysl
0.8	mot_ex1	Overall_ME	Mot_ex1
	Mot_ex2	Total_EM	Mot_ex2
	auto_leng_total	auto_leng_months	auto_leng_months
0.7	Age	Age_groups	Age_groups
	Direct_D	Span	Direct_D
	OS	OT	OT
	Total_O	OS	OS
	auto_leng_num	Total_leng_auto	auto_leng_num

**Table 6 ijerph-18-10785-t006:** Automatic selection of variables important for EF prediction.

Variables Selected in the Subset of Variables with <0.9 Pearson Correlation Tolerance between Variables	Variables Selected in the Subset of Variables with <0.8 Pearson Correlation Tolerance between Variables	Variables Selected in the Subset of Variables with <0.7 Pearson Correlation Tolerance between Variables
Imi_post	OV_compr	Overall_ME
Mem_verbal	Mem_verbal	OV_compr
Direct_D	Overall_mental_contr	Span
Constr_praxis	Direct_D	Clock_copy
Clock_copy	Constr_praxis	Const_praxis
Errors	Errors	Total_leng_auto
WV_compr	WV_compr	Ide_praxis

**Table 7 ijerph-18-10785-t007:** Automatic selection of variables important for EF prediction based on weights.

Variables with Correlation under 0.9	Weight	Variables with Correlation under 0.8	Weight	Variables with Correlation under 0.7	Weight
Constr_praxis	56	Constr_praxis	56	Constr_praxis	48
Mem_verbal	45	Mem_verbal	46	Mem_verbal	40
Direct_D	45	VW_compr	41	Clock_copy	36
OP	32	Raven	33	Mem_recog	34
Mem_recog	32	Direct_D	33	OP	33
Raven	31	Secs	33	Secs	33
Secs	31	OP	32	Raven	32
Age	28	Mem_recog	29	PPVT	32
PPVT	26	Age	27	Age	29
Total_denomin	24	PPVT	26	Spont_lang	29
Clock_copy	24	Total_denomin	26	Total_denomin	27
Verbal_flu	23	OS	26	Clock_order	26
VW_compr	22	Clock_order	24	Verb_flu	25
Clock_order	22	Verbal_flu	24	VW_compr	24
Imi_post	19	Imi_post	19	Imi_post	19
Spont_lang	19	Clock_copy	18	Ide_praxis	19
Imi_post	19	auto_leng_num	17	Overall_ME	19
auto_leng_num	17	Imi_post	17	OV_compr	17
Ide_praxis	16	Overall_EM	16	Span	17
Overall_ME	16	Ide_praxis	14	Gender	15

**Table 8 ijerph-18-10785-t008:** Comparison of performance as regards the balanced accuracy of the models used.

Model	Subset Variables Correlation <0.7			Subset Variables Correlation <0.8			Subset Variables Correlation <0.9		
Metric	Acc	F1	AUC	Acc	F1	AUC	Acc	F1	AUC
Random Forest	73.6%	68.0%	74.4%	**86.8%**	**82.5%**	**87.0%**	84.2%	81.0%	85.4%
Logistic regression	71.0%	61.5%	63.3%	73.6%	66.1%	69.6%	71.0%	67.0%	77.0%
Support vector machine	57.8%	51.0%	55.0%	55.2%	47.5%	51.6%	60.5%	58.1%	70.4%

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
