# Peer review of "Executive Functioning in Adults with Down Syndrome: Machine-Learning-Based Prediction of Inhibitory Capacity"

_ijerph, 2021, doi:10.3390/ijerph182010785_

Round 1
Reviewer 1 Report
Congratulations, this is an excellent piece of research implemented with an original methodology. I regret, however, that you have decided to deal only with one single dependent variable (inhibition capacity), even if it is particularly important in itself. Using several dependent variables related to other executive aspects would have given more scope and significance to your work for a relatively limited amount of aditional testing and measurement. May be some other time.
Author Response
We appreciate your comments and suggestions. You are right, but unfortunately, we did not collect more variables related to executive functions for this study.
We agree that it would have been interesting to include other executive functions and we had considered it for the study. However, we observed that in some tests there was a floor effect in the results, for example in the reverse digits test to evaluate working memory and it was difficult to collect results in these tests in this population.
In order to consider this idea in the manuscript, we have added this sentence in the limitations in the discussion section indicated in yellow as follows:
It would also have been interesting to include in the study more variables related to other executive aspects, apart from the inhibition capacity. However, we observed that in some other tests related to executive functions (e.g. Reverse digits), there was a floor effect in the results in relation to the cognitive profile of this population, so we decided not include these other measures.
Reviewer 2 Report
The authors leverage machine learning techniques to study key variables to predict inhibition capacity in adults with Down Syndrome. The study has public health significance, however, the machine learning methods and research process are not clearly presented. I don't think machine learning gain merits than traditional modeling or latent variable approach by reading the manuscript. I hope the authors can clearly present the research process and justify the methods.
- Line 210, is there a reference to justify that 0,1,2 is significantly different from 3,4,5 so that a dichotomized variable can be defined for the cats&dogs test? Have the authors considered other cut-off points as sensitivity analyses?
- Line 231, what does the notation in equation means? It is not clear that how the data was recovered. Why did the authors chose this method instead of simple imputation or multiple imputation?
- How did the authors make the decision to remove these variables? Some of the correlated variable could be among K best features. What are the strategies to avoid removing "best features"?
- Have the authors tried PCA? Is "remove highly correlated variables" better than PCA or other approach?
- Section 2.2.3.1 is titled as RF as variable selection algorithm but the paragraph is more about the model instead of how features are selected. Consider reorganize the paragraph.
- For 2.2.3.2, is it a univariate model process?
- Have the authors tried lasso?
- I recall the total number of participant is 118 but Table 3 shows more than 9000. Can the authors clarify?
- Table 5 is not clearly presented. OT is removed because it is highly correlated to Total and OS? This is confusing. Which group of variables are correlated with more than 0.9 Pearson Correlation?
- What method was used for table 6 feature selection? Why correlation <0.8 is chosen as the set of variables to be included in the final model in line 355?
-
What about F1 score or AUC for models in Table 8? What is the distribution of response variable and the basic demographic variables?Are all the models used the same set of variables?
Author Response
We appreciate your suggestions and we have answered each question. Regarding the application of machine learning approach, we also agree that machine learning is not the only approach that may be applied to these data in the study. We selected it because it is data driven and requires few decisions for selecting potentially predictive variables, but we are not claiming that it is superior to other methods. We selected the approach that we thought was best suited to our multivariate dataset.
We have specified in the text these sentences after figure 1, added in yellow color:
Machine learning is not the only approach that may be applied to these data in the study. We selected the approach that we thought was best suited to our multivariate dataset, because it is data driven and requires few decisions for selecting potentially predictive variables.
1. Line 210, is there a reference to justify that 0,1,2 is significantly different from 3,4,5 so that a dichotomized variable can be defined for the cats&dogs test? Have the authors considered other cut-off points as sensitivity analyses?
The variable Cats-and-Dogs was converted to a dichotomous domain assigning zero (0) to the values 0,1,2 and one (1) to the values 3,4,5 for the analysis. We decided this, taking into account that between 0-2 the performance is worse than from a value of 3.
In order to clarify it we propose add this explanation in the text, indicated in yellow color:
In the same way, the procedure to convert the variable Cats-and-Dogs from an initial polychotomous domain to a dichotomous domain is to assign zero (0) to the values 0,1,2 and one (1) to the values 3,4,5, since it is considered that between 0-2 the performance is worse than from a value of 3. This gives a binary response variable, whose behavior is determined by the binomial distribution.
2. Line 231, what does the notation in equation means? It is not clear that how the data was recovered. Why did the authors chose this method instead of simple imputation or multiple imputation?
Subsection, 2.2.1 explains in summary, how the process was carried out. On the other hand, simple imputation or multiple imputation are based on the probability density function. Our proposed methods are based on the same approach, but taking the advantage of machine learning capacity. Support vector machine is widely used to missing values imputation in several computer science approaches [1] [2] [3], where the data is very sensitive like the one used in our study.
[1] AYDILEK, Ibrahim Berkan; ARSLAN, Ahmet. A hybrid method for imputation of missing values using optimized fuzzy c-means with support vector regression and a genetic algorithm. Information Sciences, 2013, vol. 233, p. 25-35.
[2] WANG, Xian, et al. Missing value estimation for DNA microarray gene expression data by Support Vector Regression imputation and orthogonal coding scheme. BMC bioinformatics, 2006, vol. 7, no 1, p. 1-10.
[3] SIVAPRIYA, T. R.; KAMAL, A. N. B.; THAVAVEL, V. Imputation and classification of missing data using least square support vector machines–a new approach in dementia diagnosis. International journal of advanced research in artificial intelligence, 2012, vol. 1, no 4, p. 29-33.
3. How did the authors make the decision to remove these variables? Some of the correlated variable could be among K best features. What are the strategies to avoid removing "best features"?
The features to be removed are selected using the expert evaluation method. This is, in a meeting of experts, in with a committee method, where the experts, using the vote, decided which variables are more important, in terms to explain Cats-and-Dogs output.
4. Have the authors tried PCA? Is "remove highly correlated variables" better than PCA or other approach?
PCA is based on the eigenvalues approach and uses a transformation to generate new no correlated variables. Instead, our basic approach uses the normalized covariance to select non correlated variables. In this way, we can observe directly the relationship of the observed variable, in terms of other ones. This let us analyze the variables in the original domain and nature of each one. This is very important to determine which of them we can remove, based on the opinion of the experts.
5. Section 2.2.3.1 is titled as RF as variable selection algorithm but the paragraph is more about the model instead of how features are selected. Consider reorganize the paragraph.
Ok, complemented in the manuscript.
It is an ensemble of decision trees (67) through the technique called bagging in order to increase the generalization capacity and decrease the variance for the desired performance metrics, with the purpose to select, in an objective way, the variables that impact the prediction of the output Cats-and-Dogs. This approach works by calculating the entropy (68) of the data for each tree and using them to determine the variables that provide the most classification information.
6. For 2.2.3.2, is it a univariate model process?
Nno, is a multivariate model process. Complemented in the manuscript.
Please see the attachment.
“.......As we can see in Ec.4, the incidence of Xi array, of size j variables, is represented as the logarithmic probabilities related to the occurrence of the output variable Yi, which is dichotomous for this case. ……”
7. Have the authors tried lasso?
No. Although there are many regression methods in literature, we have used the most common in the machine learning area. However, it could be very interesting in future works, to explore more traditional statistics methods to compare with the results of the present here..
8. I recall the total number of participant is 118 but Table 3 shows more than 9000. Can the authors clarify?
Thank you so much. Typo error. Changed.
Table 3. Data recovered from the database for machine learning analysis
|
Description |
Amount |
|
Total complete data |
7218 |
|
Total data recovered |
114 |
|
Total data available for processing |
7332 |
|
Total variables |
39 |
|
Total registers |
188 |
9. Table 5 is not clearly presented. OT is removed because it is highly correlated to Total and OS? This is confusing. Which group of variables are correlated with more than 0.9 Pearson Correlation?
We appreciate your observation regarding this mistake. We have already changed TOTAL by OT and we have already changed it in the manuscript.
10. What method was used for table 6 feature selection? Why correlation <0.8 is chosen as the set of variables to be included in the final model in line 355?
As is described in section 3.3, the intersection (common outputs) of the three used algorithms for feature selection, is summarized in table 6. It is important to take into account that the algorithms were executed per each subset conformed for variables with lower correlation coefficient than the configured threshold. In this case 0.7, 0.8 and 0.9 respectively. Finally, the 0.8 subset of variables was chosen based on the psychologist expert's opinion, in this case, based on our experience and knowledge about the focus study area. We have selected this column because it includes more variables which represent several cognitive functions, and this column includes specifically verbal comprehension.
We have indicated it in the manuscript as follows:
We have selected the column corresponding to variables with< 0.8 Pearson correlation, based on our experience and knowledge about the focus study area, taking into account that more cognitive functions are represented in this column, and also in the results shown in table 8. So, the best variables predictors of inhibition capacity were functions related to praxis, memory, attention, executive functions and language: Oral verbal comprehension, Verbal memory, Overall mental control, Direct digits, Constructive praxis, Errors of memory of images, and Written verbal comprehension.
11. What about F1 score or AUC for models in Table 8? What is the distribution of response variable and the basic demographic variables?Are all the models used the same set of variables?
F1 Score and AUC were included. All models use the same set of variables.
Table 8. Comparison of performance as regards balanced accuracy of the models used.
|
Model |
Subset variables correlation < 0.7 |
Subset variables correlation < 0.8 |
Subset variables correlation < 0.9 |
||||||
|
Metric |
Acc |
F1 |
AUC |
Acc |
F1 |
AUC |
Acc |
F1 |
AUC |
|
Random Forest |
73.6% |
68.0% |
74.4% |
86.8% |
82.5% |
87.0% |
84.2% |
81.0% |
85.4% |
|
Logistic regression |
71.0% |
61.5% |
63.3% |
73.6% |
66.1% |
69.6% |
71.0% |
67.0% |
77.0% |
|
Support vector machine |
57.8% |
51.0% |
55.0% |
55.2% |
47.5% |
51.6% |
60.5% |
58.1% |
70.4% |
As shown in Table 8, the model with the best performance is Random Forest, with balanced accuracy of 86% for the dataset formed by all the variables except those that show a Pearson correlation above the 0.9 threshold. It is also highlighted that the balanced accuracy metric was used for this application, so the dataset shows a considerable imbalance of 86%, as regards the amount of data per class of the Cats-and-Dogs response variable.

Reviewer 3 Report
I think this manuscript possesses a very unique and a novel approach to throw light on a very critical medical problem. Here, Acosta et al applied machine learning approaches to identify the cognitive key variables related to the inhibition capacity in adults with Down syndrome. I really liked their approach as I think the findings in this manuscript will be really helpful to strategize/tackle the symptoms of DS patients in near future. Therefore, I would highly recommend publishing this article in MDPI-IJERPH. I only have one small comment for the authors that I would like them to clarify in the discussion.
(1) A lot of machine learning approaches use Artificial Neural Network (ANN) based algorithms to predict rules governing systems these days. Why didn’t the authors consider utilizing this methodology here? It could be because of the low sample size, but I would really like a comment from the authors about this. If possible, ANN would be a better/much more unbiased approach for future to include and be compared to the results of LR, SVM and RF approaches that are used here.
Author Response
Thank you so much for your comment. We have decided to use traditional algorithms to perform the feature selection based on linear and non linear approaches. Well explained algorithms, with not deep structure, let us better understand the behavior of the selected features. However, neural networks could be used for this kind of task, but the hyper parameters related to the deep structure, in terms of number of layers and number of neurons, could hide this purpose. So, using a simple linear neural network structure is the same as a linear kernel transformation, such as SVM algorithm works, since we are working only in the linear approach of this algorithm.
Reviewer 4 Report
Jojoa-Acosta and colleagues proposed a research article aimed at assessing the predictive potential of machine learning to identify cognitive key cognitive alterations in adults with Down syndrome. For this purpose, the authors developed an algorithm based on the use of different cognitive tests to collect data on the cognitive performance of a wide cohort of DS patients. Through the collection of data and their interpretation performed by the methods here developed, the authors identified praxis, memory, attention, executive functions and language as the best predictors of cognitive impairments due to aging. Overall, the manuscript is interesting, however, it is not well written and it does not fit well with the main topics of the IJERPH Journal. Below are reported some critical comments that the authors have to address:
1) Please shorten the title;
2) Please revise the section “affiliation” adding the precise Institution, City and Country of all the authors;
3) Revise the grammar: “Which cognitive domains most predict the capacity for inhibition in adults with DS is not fully known in order to optimize independent 18 living skills.”;
4) The manuscript should be extensively revised by an English native speaker;
5) The contents of the Introduction section are ok, however, they should be better described from a grammar point of view;
6) The Introduction section is too long. It should be shortened and improved;
7) Besides each test reported in Table 1, the authors have to indicate the specific reference;
8) Avoid the repetition of the word “variable” in the following sentence: “This work proposes a machine learning-based system for the automatic selection of the main variables that best explain the Cats-and-Dogs response variable (EF).”;
9) In the Methods section, the authors have to include a wide description of the statistical tests applied to the algorithm here presented;
10) The data generated in the present study are not strictly pertinent to the topic of the Journal. I suggest addressing the above revisions and consider submitting the manuscript to other journals with a specific focus on cognitive impairments.
Author Response
Thank you for all your comments and suggestions which we have taken them into account.
1) Please shorten the title;
We have changed the title following your suggestions:
Executive functioning in adults with Down's syndrome: Machine learning based prediction of inhibitory capacity
2) Please revise the section “affiliation” adding the precise Institution, City and Country of all the authors;
Thank you for your comment and sorry for the missing information, it was a mistake. We have already added all these data to the affiliation section, which is shown in yellow in the text.
3) Revise the grammar: “Which cognitive domains most predict the capacity for inhibition in adults with DS is not fully known in order to optimize independent 18 living skills.”;
Thank you for your comment. We have changed the sentence in the abstract section. We have indicated it in yellow color.
4) The manuscript should be extensively revised by an English native speaker;
All the text has been edited by an English native reviewer.
See all changes indicated in yellow color, especially in the introduction section. Changes in the specific paragraphs are indicated in the answer to the questions 6. Paragraphs modified or rewritten from the original version are indicated in yellow in the text. In the discussion section we have included few changes and we have added this sentence in the limitations paragraph, indicated in yellow as follows:
It would also have been interesting to include in the study more variables related to other executive aspects, apart from the inhibition capacity. However, we observed that in some other tests related to executive functions (e.g., Reverse digits), there was a floor effect in the results in relation to the cognitive profile of this population, so we decided not include these other measures.
5) The contents of the Introduction section are ok, however, they should be better described from a grammar point of view;
An English native reviewer has modified the Introduction section and rewritten some paragraphs in order to improve grammar. We have indicated all changes in yellow color in the manuscript.
Paragraphs modified in the introduction section are indicated in the next answer.
6) The Introduction section is too long. It should be shortened and improved;
We have shortened and improved the Introduction from the original version, including the essential content for the paper.
From the original manuscript we have modified, rewritten or deleted these paragraphs as follows and indicated in yellow color in the text:
Paragraph 1: modified and rewritten.
Paragraph 2: deleted
Paragraph 3: modified
Paragraph 4: rewritten
Paragraph 5: deleted
Paragraph 6: modified
Paragraph 7: modified
Paragraph 8: deleted
Paragraph 9: deleted
Paragraph 10: modified
Paragraph 11: modified and partially deleted at the end of the paragraph
Paragraph 12: modified
Paragraph 13: similar
7) Besides each test reported in Table 1, the authors have to indicate the specific reference;
We have included in Table 1 all the specific references of each test. Some tests have been created ad hoc for the study and we have also indicated in the table.
8) Avoid the repetition of the word “variable” in the following sentence: “This work proposes a machine learning-based system for the automatic selection of the main variables that best explain the Cats-and-Dogs response variable (EF).”; made, sentence changed for “This work proposes a machine learning-based system for the automatic selection of the main attributes that best explain the Cats-and-Dogs response variable (EF).”
9) In the Methods section, the authors have to include a wide description of the statistical tests applied to the algorithm here presented;
No statistics test, such as hypothesis test were applied to the output of the algorithms to compare the performance, since we have used punctual estimation for the metrics calculation.
10) The data generated in the present study are not strictly pertinent to the topic of the Journal. I suggest addressing the above revisions and consider submitting the manuscript to other journals with a specific focus on cognitive impairments.
This manuscript is based on cognitive issues of adults with DS. Currently, this group of population is part of public health interest. That is why we consider it adequate for the IJERPH, taking into account that the method applied for analyzing data is very original as the approach gives helpful information to the community. Moreover, nowadays aging is one of the topics that most concerns our society and, in this study, we are proposing the early detection of cognitive decline in adults with DS that are living longer and presenting early aging.
For these reasons we consider IJERPH as a very good option to publish our work. However, we could accept the suggestion to present our work into another journal, if you consider that.
Round 2
Reviewer 2 Report
Thanks for addressing my comments. The current manuscript looks good to me.
Author Response
R:/ Thank you so much.
Reviewer 4 Report
The authors have addressed some of my comments. The manuscript was significantly improved from a grammar point of view. However, still some issues should be addressed.
Although the authors have described the hypothesis tests used in their model, a statistical analysis of the data should be also described and applied, if applicable.
Author Response
R:/
The use of parametric statistical methods are high valuable for applications where the behavior of the data is Homoscedastic, Normally distributed and non-correlated. However, for our data these assumptions were not satisfied. Therefore, we had to apply non-parametric state of the art methods. In this case, we have preferred machine learning methods instead non-parametric statistical methods to the feature selection task, because its versatility and high precision mentioned in literature [1, 2, 3].
For sure, in a future work, a general comparison between results obtained using both statistical and machine learning methods, could be a good contribution to the state of the art.
[1] CHANDRASHEKAR, Girish; SAHIN, Ferat. A survey on feature selection methods. Computers & Electrical Engineering, 2014, vol. 40, no 1, p. 16-28.
[2] KHALID, Samina; KHALIL, Tehmina; NASREEN, Shamila. A survey of feature selection and feature extraction techniques in machine learning. En 2014 science and information conference. IEEE, 2014. p. 372-378.
[3] MIAO, Jianyu; NIU, Lingfeng. A survey on feature selection. Procedia Computer Science, 2016, vol. 91, p. 919-926.
Moreover, all the text has been revised again by an English native reviewer. See all changes indicated in yellow color, specially in the discussion section.